# The Effect of *Bacillus licheniformis* MH48 on Control of Foliar Fungal Diseases and Growth Promotion of *Camellia oleifera* Seedlings in the Coastal Reclaimed Land of Korea

**DOI:** 10.3390/pathogens8010006

**Published:** 2019-01-09

**Authors:** Sang-Jae Won, Jun-Hyeok Kwon, Dong-Hyun Kim, Young-Sang Ahn

**Affiliations:** 1Division of Forest Resources, Chonnam National University, Gwangju 61186, Korea; lazyno@naver.com (S.-J.W.); wg6102@naver.com (J.-H.K.); 2Department of Fire Safety Engineering, Jeonju University, Jeollabuk-do 55069, Korea; 72donghyunkim@jj.ac.kr

**Keywords:** antagonistic bacteria, foliar fungal pathogen, lytic enzyme, oil tea tree, seedling growth, coastal area

## Abstract

This study investigated the control of foliar fungal diseases and growth promotion of *Camellia oleifera* seedlings in coastal reclaimed land through the use of *Bacillus licheniformis* MH48. *B. licheniformis* MH48 can produce lytic enzymes chitinase and β-1,3-glucanase that can inhibit foliar pathogens by 37.4 to 50.5%. Nevertheless, foliar diseases appeared in the seedlings with bacterial inoculation, and their survival rate decreased because they were unable to withstand salt stress. However, *B. licheniformis* MH48 significantly increased the total nitrogen and phosphorus contents in the soils through fixing atmospheric nitrogen and solubilizing phosphorus. The growth of seedlings with bacterial inoculation increased, particularly in root dry weight, by 7.42 g plant^−1^, which was 1.7-fold greater than that of the control. *B. licheniformis* MH48 produces the phytohormone auxin, which potentially stimulates seedling root growth. *C. oleifera* seedlings significantly increased in total nitrogen content to 317.57 mg plant^−1^ and total phosphorus content to 46.86 mg plant^−1^. Our results revealed the effectiveness of *B. licheniformis* MH48 not only in the control of foliar fungal diseases but also in the growth promotion of *C. oleifera* seedlings in coastal lands.

## 1. Introduction

Tea (*Camellia sinensis*) and oil tea (*Camellia oleifera*) have been used to produce important beverages worldwide. Notably, tea has many impressive health benefits because it contains large amounts of catechins, theanine, and caffeine [1]. Although oil tea lacks these characteristic constituents compared to tea, *C. oleifera*, one of the most famous woody plants for vegetable oil production, is distributed and cultivated widely in central and southern China. *C. oleifera* also produces a variety of secondary metabolites such as saponins and vitamins with various applications [2]. *C. oleifera* not only enhances human health but also has high economic value [3]. The Korean government has begun to study the establishment of camellia oil tea plants, including *C. oleifera,* in Saemangeum coastal reclaimed lands [4,5]. Saemangeum is the largest reclamation project on the southwest coast of Korea. The site is an estuary tidal flat on the coast with the potential to create 28,300 ha of land, and the soil type is silt loam containing clay. However, soil salinity and poor fertility limit plant growth in the region [4,5]. The cultivation of *C. oleifera* in the coastal reclaimed land of Korea suffers from a lack of knowledge regarding plant development and nutrient management.

Salinity in the coastal reclaimed land soil can induce osmotic stress in plants and reduce water and nutrient uptake, consequently degrading the plants [4,6,7,8,9]. *C. oleifera* seedlings in the Saemangeum coastal reclaimed land are stressed in terms of water and nutrient uptake and are susceptible to infection by foliar fungal pathogens, resulting in defoliation and death [10]. Although the use of chemical fungicides would effectively protect the plant, it can also lead to serious environmental pollution that might threaten human health [11]. With the increasing awareness of environmental protection and food safety, biological control methods are attracting increasing public attention. Because of their biocontrol potential, the use of plant growth-promoting bacteria (PGPB) to control plant diseases has become an important and promising approach for biological control [12,13]. PGPB have been known to prevent fungal diseases by antifungal compounds [12,14] and lytic enzymes [15,16,17]. PGPB have been shown to be able to control plant diseases [14,15,16,17], but the specific biological control capability of the PGPB associated with *C. oleifera* remains unclear [18,19].

In soils from coastal reclaimed land, the application of appropriate fertilizers is necessary to boost plant development. Intensive farming to produce high yields and quality crops requires the extensive use of chemical fertilizers, but this practice has created environmental problems, including nitrogen and phosphorus surface runoff, that lead to the eutrophication of aquatic ecosystems [20]. As a result, a resurgence of interest in environmentally friendly, sustainable, and organic agricultural methods has occurred. PGPB comprise a wide variety of soil bacteria that positively affect plant growth and yield because they can produce plant hormones, fix nitrogen, and solubilize inorganic phosphate [4,5,21,22,23]. In addition, 1-aminocycolopropane-1-carboxylate (ACC) deaminase-containing PGPB can reduce the effects of environmental stresses, including salt and drought stresses [12,13].

PGPB can produce lytic enzymes, including chitinases, glucanases, proteases, and lipases, to degrade the cell walls of fungal pathogens and prevent foliar fungal diseases [15,16,17]. However, no study has yet reported on both the control of foliar diseases and the growth improvement in *C. oleifera* through the use of PGPB. On a field survey of the Saemangeum coastal reclaimed land conducted for this study, several *C. oleifera* seedlings were found dead (Figure 1D). According to the Korean Agriculture Culture Collection (KACC; Suwon, Korea), *Botrytis cinerea* has been found in leaves of dead *C. oleifera* seedlings. Foliar fungal diseases of *Camellia* spp. caused by G*lomerella cingulata*, *Pestalotia diospyri*, and *Pestalotiopsis karstenii* are major diseases in Korea. They are potential foliar fungal pathogens to *C. oleifera* seedlings. Considering the demonstrated benefits of PGPB, this study investigated the control of plant defense enzymes against foliar fungal pathogens, including *B. cinerea,* G. *cingulata, P. diospyri, and P. karstenii*, as well as the growth promotion of *C. oleifera* seedlings in the Saemangeum coastal reclaimed land of Korea through the use of *Bacillus licheniformis* MH48.

## 2. Materials and Methods

### 2.1. Antagonistic Bacteria Growth

*B. licheniformis* MH48 was isolated from experimental sites in the Saemangeum reclaimed coastal area [4,5,14]. Soil samples were subcultured on tryptone soy agar (TSA) medium at 30 °C for 24 h. The resulting pure and single colonies were inoculated again in tryptone soy broth (TSB) medium for 48 h and mixed with 50% glycerol and stored at −70 °C for further experiments.

To examine cell growth, *B. licheniformis* MH48 was cultured in broth media [0.5% urea ((NH_2_)_2_CO), 0.2% potassium phosphate monobasic (KH_2_PO_4_), 0.3% potassium chloride (KCl), 0.1% organic compost, and 0.2% sugar] at 30 °C on a rotary shaker at 140 rpm for 7 days. The number of colony-forming units (CFUs) was counted on each inoculation day for 7 days using the serial dilution technique on TSA plates.

### 2.2. Lytic Enzyme Assays

To examine chitinase and β-1,3-glucanase activity, *B. licheniformis* MH48 was cultured on medium at 30 °C for 7 days. The bacterial culture from each inoculation day was collected, centrifuged at 12,000 rpm for 10 min and used for enzyme assays. Chitinase activity was assayed following the procedure described by Lingappa and Lockwood [24]. A reaction mixture consisting of 50 µL of bacterial supernatant, 450 µL of 50 mM sodium acetate buffer (pH 5.0), and 500 µL of 0.5% colloidal chitin solution was incubated at 37 °C for 1 h. The reaction was terminated by adding 200 µL of 1N NaOH and centrifuging at 12,000 rpm for 10 min at 4 °C. The supernatant (750 µL) was mixed with 1 mL of Schales’ reagent and boiled at 100 °C in a water bath for 15 min. Absorbance was measured at 420 nm by a UV spectrophotometer (Shimadzu, Kyoto, Japan). One unit of chitinase activity was defined as the reducing activity that releases 1 µmol of N-acetylglucosamine per hour at 37 °C.

β-1,3-Glucanase activity was determined using the method described by Liang et al. [25]. A reaction mixture containing 50 µL of bacterial supernatant, 50 µL of laminarin (10 mg mL^−1^), and 400 µL of 50 mM sodium acetate buffer (pH 5.0) was incubated at 37 °C for 1 h. The reaction was stopped by adding 1.5 mL of the 3,5-dinitrosalicylic acid (DNS) reagent and boiled in a water bath for 5 min. Absorbance at 550 nm was used to determine the concentration of reducing sugars. One unit of β-1,3-glucanase activity was defined as the amount of enzyme that catalyzes the release of 1 µmol of glucose per hour at 37 °C.

### 2.3. Antagonistic Activity of B. licheniformis MH48 against Foliar Fungal Pathogens

Antagonistic activities of *B. licheniformis* MH48 were determined by the dual culture method against foliar pathogens *B. cinerea* KACC 40854, *G. cingulate* KACC 40299, *P. diospyri* KACC 44400, and *P. karstenii* KACC 44384. These pathogens are the most important agents causing foliar fungal diseases in *C. oleifera*. They were purchased from KACC. *B. licheniformis* MH48 was streaked on one side of each agar plate, and a fungal agar plug of 5-mm diameter was made using a sterile cork borer and placed on the other side of the inoculated plate. A plate inoculating the fungal pathogen alone was used as the control. Three replicates of each plate were incubated at 25 °C for 7 days, and the growth inhibition of fungal pathogens was calculated using the formula [26]: growth inhibition percentage = (*R* − *r*) / *R* × 100; where *R* is the radial growth of foliar fungal pathogens in the control plate and *r* is the radial growth of foliar fungal pathogens in the dual culture plate.

To examine the effect of *B. licheniformis* MH48 on the hyphal morphology of foliar fungal pathogens, the mycelium at the inhibition zone by *B. licheniformis* MH48 was observed under a light microscope to examine the deformation of the hyphal structure of fungal pathogens (Olympus BX41TF, Japan). All experiments for the observation of morphological mycelia were performed in triplicate.

### 2.4. Study Area and Field Experimental Conditions

Saemangeum reclaimed land is located in an estuary tidal flat that lays at the intersection of the Mangyung and Dongjin rivers (Figure 1A). This area is approximately 400 km^2^ and is, therefore, one of the largest land reclamation projects in Korean history. The experimental sites were selected at the experimental station (35°53′37″ N, 126°41′45″ E) of the National Institute of Forest Science in the Saemangeum reclaimed lands in the southwest coastal area of Korea (Figure 1A), with soils affected by salt. The soils in the study sites were fluvio-marine deposits, and the dominant soil type of the study area was silt loam with a slope of 0 to 2%. The experimental sites at the Saemangeum reclaimed land experience a temperate climate with an annual mean temperature of 13 °C [27,28]. The long-term average annual precipitation on-site is 1252 mm, approximately 54% of which falls between June and August. Reeds (*Phragmites communis*) dominate the reclaimed land, and woody plants do not grow in the area because of salt stress (Figure 1B). In the study area, the reed community was removed and replaced with the experimental site (Figure 1C).

A field experiment was conducted using a complete block design after cutting 5 m wide × 5 m long × 1 m high furrows (Figure 1C). For each block, two lysimeter plots with a 2 m width × 5 m length × 0.3 m depth were installed. In July 2014, two-year-old seedlings with a height of 30 cm (10 seedlings) were planted in the lysimeter plots. The following two treatment groups were used in the seedling experiment, with each replicated three times: (1) control without bacterial inoculation and (2) *B*. *licheniformis* MH48 inoculation. A 1-m wide buffer was installed between the lysimeter plots (Figure 1C). The study sites were filled with 30 cm of sandy soil to alleviate saline conditions and promoting plant cultivation. In addition, a shade membrane was installed to prevent deer feeding in the study areas (Figure 1C). The surface in the study area was treated with nonwoven mulch material to suppress the occurrence of weeds. A stand bar was installed alongside each seedling to prevent shaking by strong sea winds.

One month after planting, bacteria (10 L of *B. licheniformis* MH48 culture) were diluted in 10 L of water and poured into soils adjacent to the seedling roots. Control seedlings received 20 L of water and were not treated with bacteria. The bacterial inoculation application was determined based on the recommended basal chemical fertilizer application rate for *Camellia sinensis* (N: P: K = 60:20: 30 g m^−2^). Treatments were applied approximately once per month.

### 2.5. Chemical Properties in Soils and Nutrient Content in Seedlings

Soil samples in each site were taken three times (July 2014, September 2014 and March 2015) at a depth of 0 to 30 cm adjacent to *C. oleifera* seedlings to analyze the pH, total nitrogen, and total phosphorus. The soil samples were oven-dried at 105 °C for 24 h after being sifted through a 2-mm sieve.

To determine the nutrient (total nitrogen and total phosphorus) content of *C. oleifera* seedlings in the treatments, the dry weights and nutrient concentrations of the seedlings were measured in April 2015. The seedling leaves, shoots, and roots were separated and rinsed with deionized water, and their dry weights were recorded after oven drying at 65 °C for 48 h. These samples were pulverized and filtered through a 30-mesh screen and then analyzed to determine their total nitrogen and total phosphorus concentrations. Nutrient content by the *C. oleifera* seedlings was calculated using the following formula: Nutrient content (mg plant^−1^) = [dry weight (g plant^−1^) × nutrient concentration (% plant^−1^)] × 10.

The soil pH was determined with a pH electrode (Phi-560, Beckman Coulter Inc., USA) in a 1:5 soil/water suspension. The total nitrogen concentrations of the soils were determined using the Kjeldahl method [29] following wet digestion with H_2_SO_4_. The total nitrogen concentrations of the seedlings were analyzed using an elemental analyzer (Variomax CN Analyzer, Elemental, Germany) equipped with a thermal conductivity detector (TCD) after high-temperature combustion at 1200 °C with nitrogen and helium gas. The total phosphorus contents in the soils and seedlings were determined via inductively coupled plasma-optical emission spectrometry (ICP-OES) (Optima 8300, PerkinElmer, USA) after heating the samples in a microwave oven (MARS Xpress, CEM Co., USA) followed by digestion in aqua regia (hydrochloric acid:nitric acid = 3:1).

### 2.6. Analysis of C. oleifera Seedling Survival Rate

The survival rates of the seedlings were surveyed from July 2014 to April 2015, and the seedlings were considered dead when their leaves were either dried or not present. The survival rate was calculated as the percentage of surviving seedlings.

### 2.7. Statistical Analyses

All statistics were performed using the Statistical Package for the Social Sciences (SPSS) statistical software package version 21 (Armonk, NY, USA), and the results are reported as the mean ± standard deviation. Data were evaluated by *t*-test with significance considered at *p <* 0.05.

## 3. Results

### 3.1. Effect of B. licheniformis MH48 on Foliar Fungal Pathogens

#### 3.1.1. Lytic Enzyme Production

The growth of *B. licheniformis* MH48 rapidly increased at 2 days after inoculation (Figure 2). The highest growth rate of 2.97 × 10^8^ CFU mL^−1^ was observed 2 days after incubation. After that, the growth of *B. licheniformis* MH48 gradually decreased until the end of the experimental period.

Chitinase activity increased over a period of 3 days, eventually reaching a maximum value of 0.46 unit mL^−1^ (Figure 3A). Thereafter, the chitinase activity gradually decreased, and the value was stable at 6 and 7 days after inoculation. β-1,3-Glucanase activity rapidly increased at 2 days after inoculation, eventually reaching a maximum value of 5.07 unit mL^−1^ (Figure 3B). Thereafter, the enzyme activity declined sharply until 4 days after inoculation, and no enzyme activity was detected from 5 days after inoculation to the end of the experimental period.

#### 3.1.2. Antagonistic Activity against Foliar Fungal Pathogens

The antagonistic activities of *B. licheniformis* MH48 against foliar fungal pathogens, including *B. cinerea*, *G*. *cingulata*, *P. diospyri*, and *P. karstenii*, were tested on PDA medium using the dual culture method (Figure 4), with the highest rate of inhibition (50.5%) against *P. karstenii* and the lowest (37.4%) against *B. cinerea*. Moreover, 39.9% and 38.5% of mycelial growth inhibition were observed against *G*. *cingulate* and *P. diospyri*, respectively (Figure 4).

Microscopic examination indicated that compared to controls without inoculation of *B. licheniformis* MH48, which showed normal hyphal structures, the hyphal morphologies of inhibited areas by *B. licheniformis* MH48 were abnormal with degradation, deformation, and lysis (Figure 5).

### 3.2. Effect of B. licheniformis MH48 on Growth Promotion of C. oleifera Seedlings

#### 3.2.1. Chemical Properties in Soils

The soil pH for the planted *C. oleifera* seedlings ranged from 7.05 to 7.68 with *B. licheniformis* MH48 inoculation and from 6.10 to 6.53 in the control without bacterial inoculation (Table 1). The soil pH values with the bacterial inoculation were significantly higher than those of the control.

The average content of total nitrogen in the soils of the growing *C. oleifera* seedlings ranged from 0.47 g kg^−1^ in the control to 1.35 g kg^−1^ with bacterial inoculation and the total phosphorus contents in the soils of the planted seedlings were 0.21 g kg^−1^ in the control and 2.48 g kg^−1^ with bacterial inoculation (Table 1). In the soils of the planted seedlings, the total nitrogen and total phosphorus contents with bacterial inoculation were significantly higher than those in the control (Table 1).

#### 3.2.2. Dry Weight, Nutrient Concentration and Nutrient Content of *C. oleifera* Seedlings

The leaf, shoot, and root dry weights of *C. oleifera* seedlings with *B. licheniformis* MH48 inoculation were significantly higher than those of the control seedlings (Table 2). In particular, the leaf dry weights of *C. oleifera* seedlings with bacterial inoculation increased by 2.7 times. Therefore, the bacterial inoculation had a significant effect on the growth of the seedlings.

The total nitrogen concentrations of *C. oleifera* seedlings with the bacterial inoculation were significantly higher than those of the control, except for the concentrations in leaves (Table 2). However, the total phosphorus concentrations of *C. oleifera* seedlings with bacterial inoculation were not significantly (*p* > 0.05) different from those in the control (Table 2).

The average total nitrogen contents of *C. oleifera* seedlings with the bacterial inoculation and in the control were 317.57 and 112.95 g plant^−1^, respectively (Table 2). The average total phosphorus contents of *C. oleifera* seedlings with the bacterial inoculation and in the control were 46.86 and 20.84 g plant^−1^, respectively (Table 2). The contents of total nitrogen and total phosphorus in *C. oleifera* seedlings with the bacterial inoculation were significantly higher than those in the control (Table 2).

#### 3.2.3. Survival Rate of *C. oleifera* Seedlings

In April 2015, the average survival rates of *C. oleifera* seedlings inoculated with *B. licheniformis* MH48 was 80.0%, compared to 63.3% for control (Figure 6); however, the difference in survival rates was not statistically significant (*p* > 0.05).

## 4. Discussion

Biocontrol agents play a role in promoting plant growth by suppressing phytopathogens [12,15,16]. Maung et al. [17] demonstrated the effectiveness of PGPB such as *Bacillus* spp. in reducing the prevalence and intensity of fungal disease, leading to growth improvements in plants. In this study, we focused on (1) the biological control of the foliar fungal pathogens *B. cinerea*, *G*. *cingulata*, *P. diospyri*, and *P. karstenii*, and (2) the growth promotion in *C. oleifera* seedlings in the coastal reclaimed land through the introduction of microbial antagonists such as *B. licheniformis* MH48.

### 4.1. Antagonistic Activity of B. licheniformis MH48 against Foliar Fungal Pathogens

Plant cultivation is threatened by the emergence of fungicide-resistant plant pathogens, which has resulted from the excessive application of fungicides. Moreover, the adverse effects of chemical fungicides on human health and the environment limit their usefulness [11]. Biological control of plant diseases using antagonistic bacteria is an increasingly important aspect of integrated disease control strategies in plant cultivation. Fungal cell wall-degrading enzymes such as chitinase and β-1,3-glucanase produced by antagonistic bacteria have been found to play key roles in the suppression of these phytopathogens [12,15,16,17,30]. *B. licheniformis* MH48 was found to inhibit foliar fungal pathogens, including *B. cinerea*, *G*. *cingulata*, *P. diospyri*, and *P. karstenii* (Figure 4 and Figure 5), because *B. licheniformis* MH48 produces fungal cell wall-degrading enzymes, such as chitinase and β-1,3-glucanase (Figure 3). The hyphae of foliar fungal pathogens showed mycelial abnormalities such as degradation, deformation, and lysis (Figure 5). In addition, *B. licheniformis* MH48 can produce the antifungal compound benzoic acid [14]. Benzoic acid showed antifungal activity against plant pathogens *Rhizobacteria solani* and *Colletotrichum gloeosporioides* with a minimum inhibitory concentration of 128 μg mL^−1^ against mycelial growth. Benzoic acid concentrations above 100 μg/mL degraded *R. solani* mycelia. However, benzoic acid can be obtained in a reduced amount of 3.3 g in 40 L of bacterial culture [14]. Therefore, benzoic acid from *B. licheniformis* MH48 is not likely to possess antifungal effects under field conditions.

The highest cell growth rate of *B. licheniformis* MH48 was observed at 2 days after incubation, coinciding with the maximum activity of β-1,3-glucanase (Figure 2 and Figure 3B). However, chitinase activity reached the maximum at 3 days after incubation (Figure 3A). Generally, lytic enzyme produced by antagonistic bacteria does not coincide with bacteria growth [31,32]. In particular, chitinase activity tends to slowly increase a few days earlier than β-1,3-glucanase activity [32].

In the reclaimed coastal lands of this study area, salts accumulate near the soil surface through capillary rise from the water table (Figure 1D) due to increased evapotranspiration in the surface layer [4,5,33,34]. The soil containing salt may have resulted from the dry surface soil conditions because of the low amount of precipitation that continued during the fall season in 2014 and spring season in 2015 [27,28], resulting in the capillary rise of salt from the water table (Figure 1D). Salt stress results in enhanced susceptibility to foliar fungal diseases in plants [10]. Although foliar diseases including leaf wilt and leaf spot appeared at the same time in all treatments of field experiment sites (Figure 1D), the rates of disease incidence were higher in control seedlings than in seedlings treated with *B. licheniformis* MH48 inoculation (Table 2). *B. licheniformis* MH48 appears to suppress foliar diseases through lytic enzymes (Figure 3, Figure 4 and Figure 5). In addition, plant defense-related enzymes, including chitinase and β-1,3-glucanase, accumulate in plants, which contribute to the induction of resistance in plants [15,17]. The leaf yields of *C. oleifera* seedlings with bacterial inoculation increased by 2.7 times compared to those of control seedlings (Table 2). Several more recent studies have demonstrated the biological control of fungal diseases caused by fungal pathogens through the use of effective antagonistic bacterial strains [14,15,16,17,30]. Based on these results, lytic enzymes (Figure 3, Figure 4 and Figure 5) produced by *B. licheniformis* MH48 clearly showed inhibitory effects on the growth of foliar fungal pathogenic *B. cinerea*, *G*. *cingulata*, *P. diospyri*, and *P. karstenii*. These results indicate that *B. licheniformis* MH48 may be a potential biological control agent for the management of various fungal pathogens.

### 4.2. Growth Promotion of C. oleifera Seedlings by B. licheniformis MH48

Salt stress can restrict plant growth because of the limited uptake of water and nutrients [4,5,35,36], which often leads to co-occurrence with infection by fungal pathogens [10]. However, *B. licheniformis* MH48 produces auxin [4,5], which can reduce salt stress [37]. Auxin reduces the levels of plant ethylene, which is a salt stress-causing substance, by synthesizing the immediate precursor of the phytohormone ethylene, 1-aminocyclopropane-1-carboxylated (ACC) deaminase [13,23]. Nonetheless, salt stress limited the survival rates of *C. oleifera* seedlings inoculated with *B. licheniformis* MH48 (Figure 6). The survival rates of *C. oleifera* seedlings did not significantly differ between bacterial inoculation and control treatments under salt stress conditions (Figure 6). *B. licheniformis* MH48 did little to effectively prevent the death of seedlings caused by salt stress.

However, *B. licheniformis* MH48 significantly increased the total nitrogen and total phosphorus contents of the soils (increases of 2.9 and 11.8-fold, respectively) compared to those of the control (Table 1) because of atmospheric nitrogen fixation and phosphorus release [4,5,12,38,39]. Specifically, several PGPB including *Bacillus* spp. were able to release significant amounts of useful minerals including phosphorus from rocks [38,39]. The accelerated breakdown of rock by plants or associated biological processes, in contrast with chemical and physical break-down, can be partly attributed to the solubilizing activity of PGPB that can colonize plant roots and organic acids exuded by roots. The valuable pH range for the uptake of nitrogen and phosphorus in plants is from pH 7.0 to 7.5, as values in this range induce increases in plant growth [37]. In this study, the soil pH increased significantly to within a range of 7.05 to 7.26 in the bacterial inoculation treatment (Table 1), which led to significant increases in the nutrient content and growth in seedlings (Table 2); this occurred because PGPB elude soil acidification by increasing the pH and producing capsular envelopes to protect themselves [16]. In addition, the auxin secreted by the bacteria can also promote root development and stimulate the formation of lateral roots and absorbent root hairs [4], resulting in increased nutrient content and growth of *C. oleifera* seedlings (Table 2). Our previously study [5] showed that *B. licheniformis* MH48 can lead to auxin accumulation in field soil. The optimal level of auxin for supporting root growth is very low, approximately 5 orders of magnitude lower than that for shoots [12]. Park et al. [4] have shown that *B. licheniformis* MH48 PGPB can stimulate *C. japonica* seedling development, including nutrient content and yields in coastal areas under salt stress conditions, consistent with results of our previous studies. The results of the present study suggest that inoculation with *B. licheniformis* MH48 resulted in an increase in the nutrient content and growth of *C. oleifera* seedlings under salt stress in coastal reclaimed land. Therefore, *B. licheniformis* MH48 may be a potential regulator of *C. oleifera* seedling cultivation in sustainable and ecological cultivation systems in reclaimed coastal lands.

## 5. Conclusions

*B. licheniformis* MH48 can prevent dissemination and lower virulence of fungal pathogens by producing lytic enzymes (Figure 3). It can also improve nutrient content and growth of *C. oleifera* seedlings in coastal areas under salt stress conditions (Table 2). Results of the present study together with those of previous work, provide strong evidence that *B. licheniformis* MH48 is not only an effective biocontrol agent of foliar fungal diseases but also beneficial for the growth of *C. oleifera* seedlings.

## Figures and Tables

**Figure 1 pathogens-08-00006-f001:**
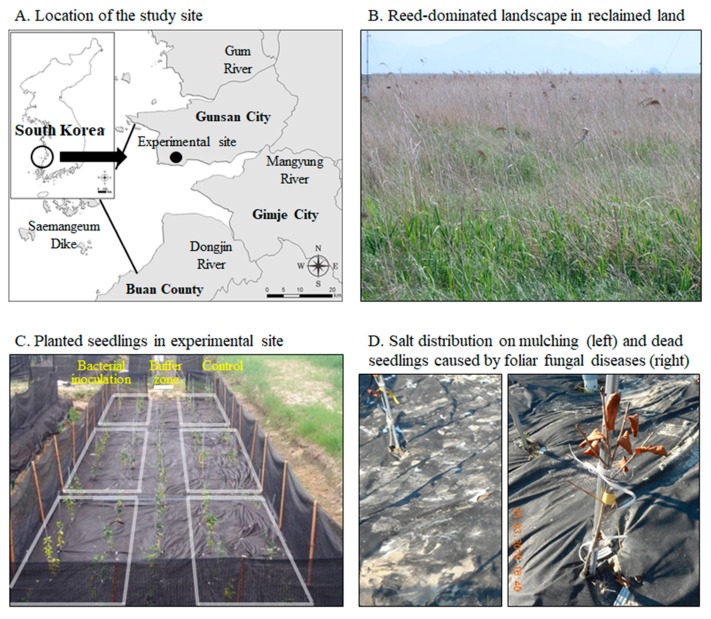
Location of the study sites (**A**); reed-dominated landscape in reclaimed land (**B**); planted seedlings of *Camellia oleifera* in the experimental site (**C**); and salt distribution on mulching (left) and dead seedlings caused by foliar fungal diseases (right) (**D**) in the Saemangeum reclaimed land.

**Figure 2 pathogens-08-00006-f002:**
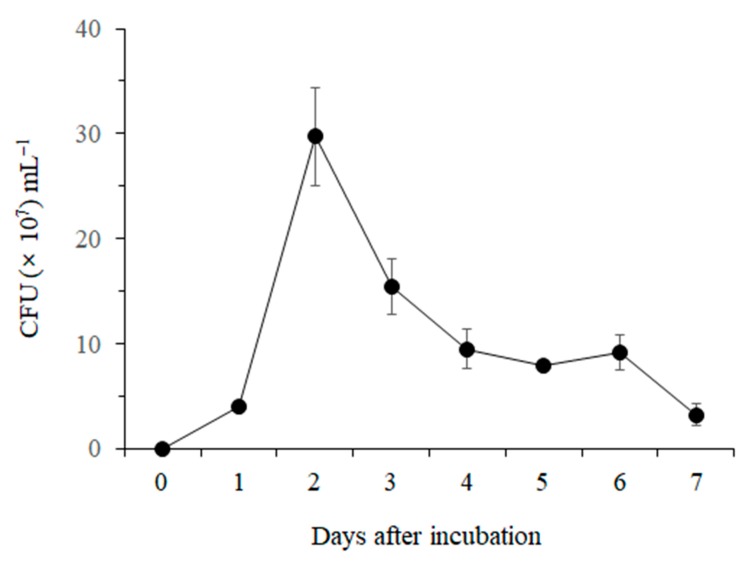
Cell growth curves of *B. licheniformis* MH48.

**Figure 3 pathogens-08-00006-f003:**
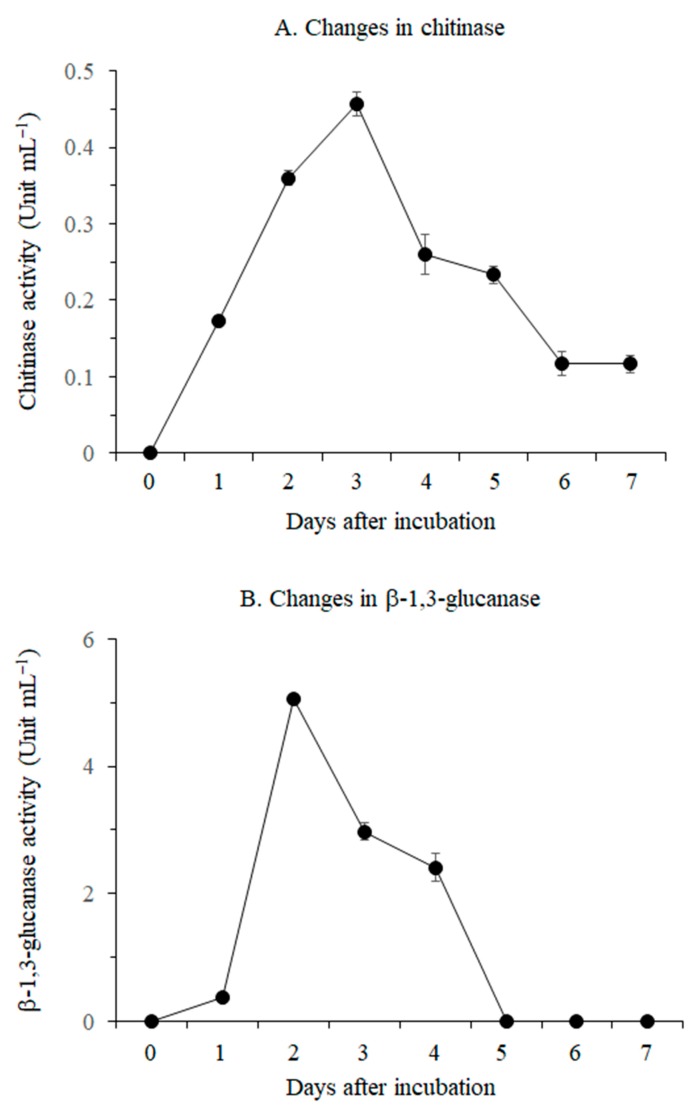
Changes in chitinase (A) and β-1,3-glucanase (B) activities in the medium after incubation of *B. licheniformis* MH48.

**Figure 4 pathogens-08-00006-f004:**
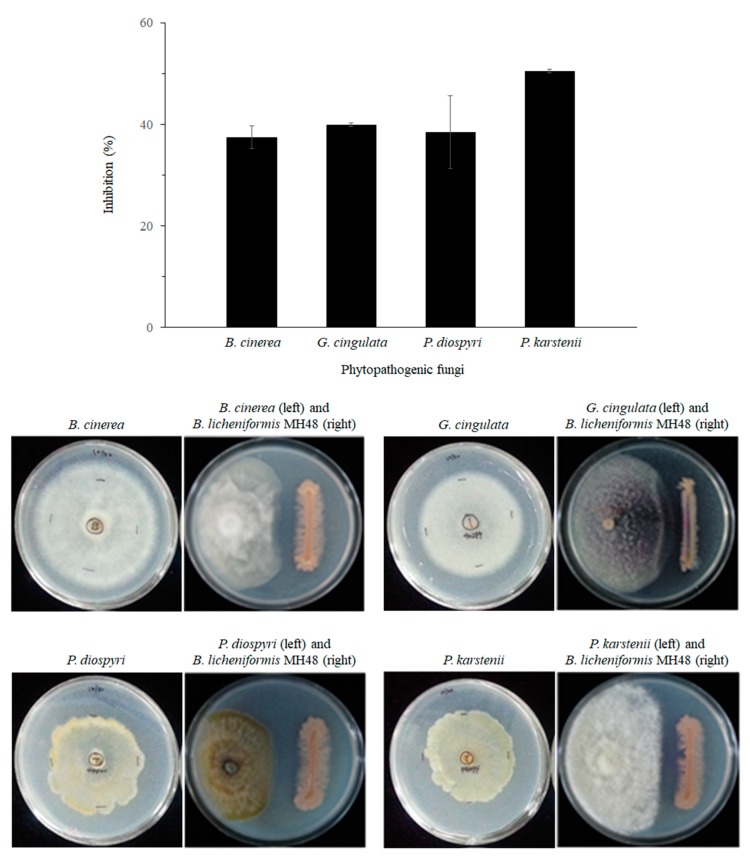
Antagonistic activities of *B. licheniformis* MH48 against *B. cinerea*, *G*. *cingulata*, *P. diospyri*, and *P. karstenii* based on the dual culture method.

**Figure 5 pathogens-08-00006-f005:**
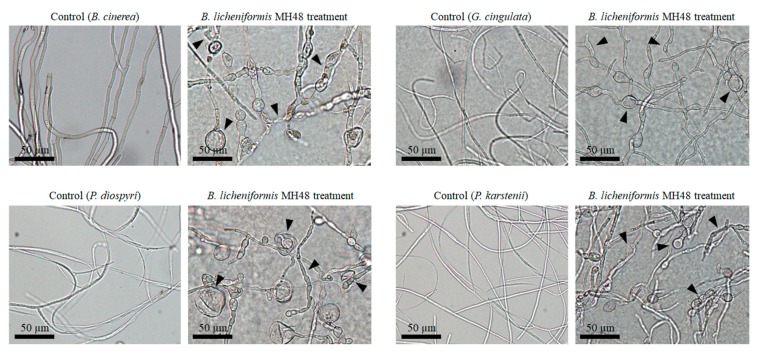
Light microscopy examination to determine effects of lytic enzymes on hyphal morphologies of *B. cinerea*, *G*. *cingulata*, *P. diospyri*, and *P. karstenii* incubated with *B. licheniformis* MH48.

**Figure 6 pathogens-08-00006-f006:**
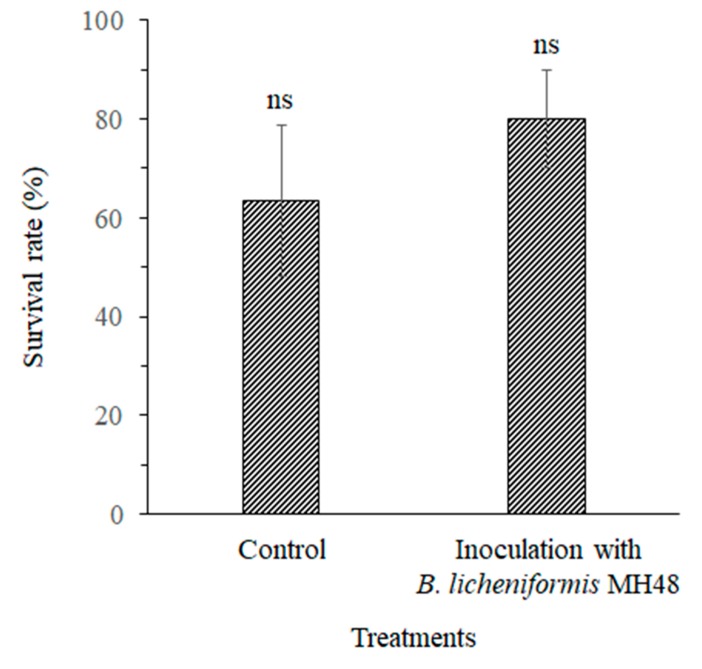
Average survival rates of *C. oleifera* seedlings in control and inoculation with *B. licheniformis* MH48 treatments in reclaimed coastal land. ns indicates a nonsignificant difference between variables at *p* < 0.05.

**Table 1 pathogens-08-00006-t001:** The pH, total nitrogen, and total phosphorus in the soils for the planted *C. oleifera* seedlings of the control and treatments with *B. licheniformis* MH48 inoculation in reclaimed coastal land.

	Control	Inoculation with *B. licheniformis* MH48
pH	6.30 ± 0.15 *	7.29 ± 0.21 *
Total nitrogen (g kg^−^^1^)	0.47 ± 0.21 *	1.35 ± 0.57 *
Total phosphorus (g kg^−^^1^)	0.21 ± 0.07 *	2.48 ± 0.93 *

* Indicates a significant difference between variables at *p* < 0.05.

**Table 2 pathogens-08-00006-t002:** Dry weight, nutrient concentration, and nutrient content of *C. oleifera* seedlings of control and inoculation with *B. licheniformis* MH48 treatments in coastal reclaimed land.

	Control	Inoculation with *B. licheniformis* MH48
Dry weight (g plant^−^^1^)		
Leaf	0.91 ± 0.23 *	2.46 ± 1.00 *
Shoot	5.25 ± 1.51 *	10.07 ± 4.60 *
Root	4.45 ± 0.83 *	7.42 ± 0.93 *
Total nitrogen concentration (% plant^−^^1^)		
Leaf	1.57 ± 0.18 *	2.30 ± 0.37 *
Shoot	0.94 ± 0.10	1.28 ± 0.28
Root	1.09 ± 0.13 *	1.72 ± 0.32 *
Total phosphorus concentration (% plant^−^^1^)		
Leaf	0.17 ± 0.03	0.21 ± 0.05
Shoot	0.21 ± 0.05	0.24 ± 0.04
Root	0.19 ± 0.03	0.21 ± 0.03
Nutrient content (mg plant^−^^1^)		
Total nitrogen	112.95 ± 18.11 *	317.57 ± 126.71 *
Total phosphorus	20.84 ± 3.11 *	46.86 ± 13.11 *

* Indicates a significant difference between variables at *p* < 0.05.

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
