# Peer review of "The Effect of Bacillus licheniformis MH48 on Control of Foliar Fungal Diseases and Growth Promotion of Camellia oleifera Seedlings in the Coastal Reclaimed Land of Korea"

_pathogens, 2019, doi:10.3390/pathogens8010006_

Reviewer 1 Report

The manuscript describes the effect of Bacillus licheniformis on foliar fungal species and growth promotion of Camellia oleifera. The results show that B. licheniformis inhibits growth of B. cinerea, G. cingulata, P. diospyri and P. karstenii, and promotes the growth of C. oleifera seedlings.

The percentages presented in the manuscript are too significant. For instance, in page 2, line 6, change 37.43 and 50.50 to 37.4 and 50.5, respectively. Check the entire manuscript for this and change accordingly.

Page 10, lines 9-10: This sentence does not belong here. Move it to line 12.

Figure 3A: Chitinase activity of B. licheniformis reached the maximum on Day 3 whereas its growth was peaked on Day 2. Discuss/Explain.

Figure 4: Use the full fungal names in the legend. Also, label the plates appropriately. It seems that the fungal species and B. licheniformis are growing together on each plate on the left. Current labels make the readers think that left plates are just B. licheniformis.

Figure 5: There is no scale bars in the micrographs. Also, it would be great if authors indicate the deformation of the hyphae using arrows.

Table 1: Authors need to indicate that this is the soil pH for the planted C. oleifera seedlings in the title.

Author Response

Response to Reviewer 1 Comments

I appreciate your valuable time and critical comments. I agree with your suggestions and comments. Therefore, I have revised the manuscript according to your suggestions. The revised sentences are highlighted in yellow in the revised manuscript.

*********************

The manuscript describes the effect of Bacillus licheniformis on foliar fungal species and growth promotion of Camellia oleifera. The results show that B. licheniformis inhibits growth of B. cinerea, G. cingulata, P. diospyri and P. karstenii, and promotes the growth of C. oleifera seedlings.

- The percentages presented in the manuscript are too significant. For instance, in page 2, line 6, change 37.43 and 50.50 to 37.4 and 50.5, respectively. Check the entire manuscript for this and change accordingly.

The percentages in the manuscript have been changed to have one decimal place. (Line 5-6 in Page 2 and Line 21 in Page 12)

- Page 10, lines 9-10: This sentence does not belong here. Move it to line 12.

► I have revised it accordingly. (Line 13-15 in Page 10)

- Figure 3A: Chitinase activity of B. licheniformis reached the maximum on Day 3 whereas its growth was peaked on Day 2. Discuss/Explain.

Generally, lytic enzyme produced by antagonistic bacteria does not coincide with bacteria growth (Jeon, 2017). I have inserted the discussion as follows;

The highest cell growth of B. licheniformis MH48 was observed at 2 days after incubation, coinciding with the maximum activity of β-1,3-glucanase (Figures 2 and 3B). However, chitinase activity reached the maximum at 3 days after incubation (Figure 3A). Generally, lytic enzyme produced by antagonistic bacteria does not coincide with bacteria growth [32;33]. In particular, chitinase activity tends to slowly increase a few days earlier than β-1,3-glucanase activity [33]. (Line 7-12 in Page 14)

- Figure 4: Use the full fungal names in the legend. Also, label the plates appropriately. It seems that the fungal species and B. licheniformis are growing together on each plate on the left. Current labels make the readers think that left plates are just B. licheniformis.

I have used full fungal names in the legend. I also inserted a label on each plate in Figure 4. (Page 29)

- Figure 5: There is no scale bars in the micrographs. Also, it would be great if authors indicate the deformation of the hyphae using arrows.

I have inserted scale bars in the micrographs. I also indicated the deformation of hyphae using arrows in Figure 5. (Page 30)

- Table 1: Authors need to indicate that this is the soil pH for the planted C. oleifera seedlings in the title.

► I have revised as follows;

Table 1. The pH, total nitrogen, and total phosphorus in the soils for the planted C. oleifera seedlings of the control and treatments with B. licheniformis MH48 inoculation in reclaimed coastal land. (Line 3-5 in Page 24)

Reviewer 2 Report

The paper contains very interesting results, but it needs some modifications, before publication.

more precisely:

In the literature the use of the following definition:” Plant Growth Promoting Bacteria” (PGPB), instead of PGPR is now preferred (see the papers of B. Glick, University of Waterloo). 

Line 13 Not all PGPB produce  1-aminocycolopropane-1-carboxylate (ACC) deaminase!

Line 19. This sentence has been partly already written in line 13

.

 Pag.6,line 16: please write: “against the foliar pathogens B. cinerea, G. cingulata, P. diospyri, and P. karstenii, that are the most important agents causing foliar fungal diseases in C. oleifera” [3].

Please delete the first part of the conclusions, already written in the Introduction

Anyway, the most important point is the following:

have the authors determined the soil  P soluble fraction or total soil P content as written in Table 1? If the total P content has been determined, it is not clear to understand how bacteria producing P solubilizing enzymes could increase soil P content.

Author Response

Response to Reviewer 1 Comments

I appreciate your valuable time and critical comments. I agree with your suggestions and comments. Therefore, I have revised the manuscript according to your suggestions. The revised sentences are highlighted in yellow in the revised manuscript.

*********************

The paper contains very interesting results, but it needs some modifications, before publication.

more precisely:

- In the literature the use of the following definition:” Plant Growth Promoting Bacteria” (PGPB), instead of PGPR is now preferred (see the papers of B. Glick, University of Waterloo). 

I have changed ‘PGPR’ to ‘PGPB’ in this manuscript.

- Line 13 Not all PGPB produce 1-aminocycolopropane-1-carboxylate (ACC) deaminase!

I have revised to ‘1-aminocycolopropane-1-carboxylate (ACC) deaminase-containing PGPB. (Line 14-16 in Page 4)

- Line 19. This sentence has been partly already written in line 13.

► I have deleted it. (Line 19 in Page 4)

- Pag.6, line 16: please write: “against the foliar pathogens B. cinerea, G. cingulata, P. diospyri, and P. karstenii, that are the most important agents causing foliar fungal diseases in C. oleifera” [3].

► I have revised it. (Line 19-23 in Page 6)

- Please delete the first part of the conclusions, already written in the Introduction

► I have deleted the first part of the conclusion. (Line 1 in Page 17)

Anyway, the most important point is the following:

- have the authors determined the soil P soluble fraction or total soil P content as written in Table 1? If the total P content has been determined, it is not clear to understand how bacteria producing P solubilizing enzymes could increase soil P content.

Several bacteria were able to release significant amounts of useful minerals including phosphorus from rocks (Puente et al., 2004a; b). I have revised it as follows;

However, B. licheniformis MH48 significantly increased the total nitrogen and total phosphorus contents of the soils (increases of 2.9 and 11.8-fold, respectively) compared to those of the control (Table 1) because of atmospheric nitrogen fixation and phosphorus release [4,5,13,39,40]. Especially, several PGPB including Bacillus spp. were able to release significant amounts of useful minerals including phosphorus from rocks [39,40]. The accelerated breakdown of rock by plants or associated biological processes, in contrast with chemical and physical break-down, can be partly attributed to the solubilizing activity of PGPB that can colonize plant roots and organic acids exuded by roots. (Line 21 in Page 15 to Line 4 in Page 16)

Reviewer 3 Report

The manuscript - “The Effect of Bacillus licheniformis MH48 on Control of Foliar Fungal Diseases and Growth Promotion of Camellia oleifera Seedlings in the Coastal Reclaimed Land of Korea, evaluates the possibility of controlling the foliar fungal diseases and growth promotion of Camellia oleifera seedlings through the use of a bacteria, Bacillus licheniformis MH48. The authors concluded that the antagonistic bacteria - B. licheniformis MH48 - controlled the foliar fungal diseases and promoted the growth of C. oleifera seedlings.

In fact, plant fungal diseases are an important and growing problem in agriculture, especially for a tree so significant as the C. oleifera. The search for alternative treatments are important, so this report has relevant results for the scientific community.  

However, the manuscript has some points, that need to be adjusted before publication.

General points:

-        An English review is needed, for there are some grammatical typos along the text;

-        The text needs to be formatted along the manuscript;

Introduction:

-        Why choose specifically B. cinerea, G. cingulata, P. diospyri, and P. karstenii? This should be better clarified;

-        In the part of the antioxidant activity and bioactive compounds of tea, this reviewer advises the checking of these two reports to complete the information: doi: 10.1186/s12870-015-0574-6 and doi: 10.2174/09298673113209990158.

Material and Methods:

-        Indicate all strains used (including the considered pathogens);

-        How were the strains (all of them) identified? Chromogenic media and/or molecular method (e.g. PCR)? Describe and indicate the references;

-        Why it was not used a reference strain of Bacillus licheniformis?

Results:

-        The tables and graphs need more quality;

-        Legend pf Figure 5 is not right regarding the name of the species;

Discussion:

-        The manuscript claims that benzoic acid is responsible for the antifungal effect on the seedlings of the experimental. But, in fact, this compound has not been evaluated and its concentration determined on the trees with the bacteria. This conclusion seems to be taken only by previous results. Either a concentration of benzoic acid has to be performed or this part needs to be rewritten, in order to reserve this fact to the readers. The same happens with auxin. Reformulate the discussion on these parts;

-        The conclusion seems to have information that belongs to the discussion. This reviewer advises a correction and a condensation of the conclusion.

Author Response

Response to Reviewer 3 Comments

I appreciate your valuable time and critical comments. I agree with your suggestions and comments. Therefore, I have revised the manuscript according to your suggestions. The revised sentences are highlighted in yellow in the revised manuscript.

*********************

The manuscript - “The Effect of Bacillus licheniformis MH48 on Control of Foliar Fungal Diseases and Growth Promotion of Camellia oleifera Seedlings in the Coastal Reclaimed Land of Korea”, evaluates the possibility of controlling the foliar fungal diseases and growth promotion of Camellia oleifera seedlings through the use of a bacteria, Bacillus licheniformis MH48. The authors concluded that the antagonistic bacteria - B. licheniformis MH48 - controlled the foliar fungal diseases and promoted the growth of C. oleifera seedlings.

In fact, plant fungal diseases are an important and growing problem in agriculture, especially for a tree so significant as the C. oleifera. The search for alternative treatments are important, so this report has relevant results for the scientific community.  

However, the manuscript has some points, that need to be adjusted before publication.

General points:

- An English review is needed, for there are some grammatical typos along the text;

I have performed an English review.

- The text needs to be formatted along the manuscript;

I have formatted it.

Introduction:

- Why choose specifically B. cinerea, G. cingulata, P. diospyri, and P. karstenii? This should be better clarified;

► I have explained it as follows;

During a field survey in this study, leaves of C. oleifera seedlings were blight and died in the Saemangeum coastal reclaimed land (Figure 1D). According to the Korean Agriculture Culture Collection (KACC; Suwon, Korea), Botrytis cinerea has been found in leaves of dead C. oleifera seedling. Foliar fungal diseases of Camellia spp. by Glomerella cingulata, Pestalotia diospyri, and Pestalotiopsis karstenii are major diseases in Korea. They are potential foliar fungal pathogens to C. oleifera seedlings. (Line 20 in Page 4 to Line 2 in Page 5)

- In the part of the antioxidant activity and bioactive compounds of tea, this reviewer advises the checking of these two reports to complete the information: doi: 10.1186/s12870-015-0574-6 and doi: 10.2174/09298673113209990158.

I have reviewed these two papers and revised the manuscript as follows;

Tea (Camellia sinensis) and oil tea (Camellia oleifera) belonging to genus Camellia have been used to produce important beverages worldwide. Nutritional value and healthful properties of tea are closely related to large amounts of catechins, theanine, and caffeine [1]. Although oil tea lacks these characteristic constituents compared to tea, C. oleifera, one of the most famous woody plants for vegetable oil production, is distributed and cultivated widely in central and southern China. C. oleifera also produces a variety of secondary metabolites such as saponins and vitamins with various applications [2]. (Line 2-8 in Page 3)

Material and Methods:

- Indicate all strains used (including the considered pathogens);

► I have indicated all used pathogen strains as follows;

Antagonistic activities of B. licheniformis MH48 were determined by the dual culture method against foliar pathogens B. cinerea KACC 40854, G. cingulate KACC 40299, P. diospyri KACC 44400, and P. karstenii KACC 44384. These pathogens are the most important agents causing foliar fungal diseases in C. oleifera. They were purchased from KACC. (Line 19-23 in Page 6)

- How were the strains (all of them) identified? Chromogenic media and/or molecular method (e.g. PCR)? Describe and indicate the references;

Foliar pathogens were purchased from the Korean Agriculture Culture Collection (KACC; Suwon, Korea). (Line 22-23 in Page 6)

- Why it was not used a reference strain of Bacillus licheniformis?

The objective of this study was to investigate the control of foliar fungal diseases and growth promotion of C. oleifera seedlings in coastal reclaimed land through the use of B. licheniformis MH48. (Line 2-6 in Page 5)

Results:

- The tables and graphs need more quality;

I have modified the resolution to 2000 dpi for tables and graphs.

- Legend pf Figure 5 is not right regarding the name of the species;

► I have revised the legend for Figure 5. (Page 30)

Discussion:

- The manuscript claims that benzoic acid is responsible for the antifungal effect on the seedlings of the experimental. But, in fact, this compound has not been evaluated and its concentration determined on the trees with the bacteria. This conclusion seems to be taken only by previous results. Either a concentration of benzoic acid has to be performed or this part needs to be rewritten, in order to reserve this fact to the readers. The same happens with auxin. Reformulate the discussion on these parts;

Benzoic acid: I have revised benzoic acid’s antifungal effects in this manuscript. Benzoic acid shows antifungal activity against plant pathogens with minimum inhibitory concentration of 128 μg mL-1 against mycelial growth. However, benzoic acid has reduced amount of 3.3 g in 40 L of bacterial culture. Therefore, benzoic acid from B. licheniformis MH48 is not likely to exhibit antifungal effects under field conditions. (Line 23 in Page 13 to Line 6 in Page 14)

Auxin: In a previous study, we have found that B. licheniformis MH48 can lead to auxin accumulation in field soil. The optimal level of auxin for supporting root growth is very low, approximately 5 orders of magnitude lower than that for shoots. Park et al. [4] have explained that B. licheniformis MH48 PGPB can stimulate Camellia japonica seedling development, including nutrient content and yields in coastal areas under salt stress conditions. Therefore, auxin produced by B. licheniformis MH48 seems to promote root development of C. oleifera seedlings under field conditions. (Line 13-18 in Page 16)

- The conclusion seems to have information that belongs to the discussion. This reviewer advises a correction and a condensation of the conclusion.

I have rewritten the Conclusions section (Line 1-6 in Page 17)
